# Comprehensive Strategy for Identifying Extracellular Vesicle Surface Proteins as Biomarkers for Non-Alcoholic Fatty Liver Disease

**DOI:** 10.3390/ijms241713326

**Published:** 2023-08-28

**Authors:** Nahuel Aquiles Garcia, Maiken Mellergaard, Hernan Gonzalez-King, Carlos Salomon, Aase Handberg

**Affiliations:** 1GECORP, Av. Juan Manuel de Rosas 899, Monte, Buenos Aires 7220, Argentina; gecorpinfo@gmail.com; 2Department of Clinical Biochemistry, Aalborg University Hospital, Aalborg Hobrovej 18-22, 9000 Aalborg, Denmark; 3Department of Clinical Medicine, The Faculty of Medicine, Aalborg University, 9000 Aalborg, Denmark; 4Research and Early Development, Cardiovascular, Renal and Metabolism (CVRM), BioPharmaceuticals R&D, AstraZeneca, 431 50 Gothenburg, Sweden; 5Translational Extracellular Vesicles in Obstetrics and Gynae-Oncology Group, University of Queensland, Brisbane, QLD 4029, Australia

**Keywords:** non-alcoholic fatty liver, extracellular vesicles, biomarkers, surface proteins

## Abstract

Non-alcoholic fatty liver disease (NAFLD) is a liver disorder that has become a global health concern due to its increasing prevalence. There is a need for reliable biomarkers to aid in the diagnosis and prognosis of NAFLD. Extracellular vesicles (EVs) are promising candidates in biomarker discovery, as they carry proteins that reflect the pathophysiological state of the liver. In this review, we developed a list of EV proteins that could be used as diagnostic biomarkers for NAFLD. We employed a multi-step strategy that involved reviewing and comparing various sources of information. Firstly, we reviewed papers that have studied EVs proteins as biomarkers in NAFLD and papers that have studied circulating proteins as biomarkers in NAFLD. To further identify potential candidates, we utilized the EV database Vesiclepedia.org to qualify each protein. Finally, we consulted the Human Protein Atlas to search for candidates’ localization, focusing on membrane proteins. By integrating these sources of information, we developed a comprehensive list of potential EVs membrane protein biomarkers that could aid in diagnosing and monitoring NAFLD. In conclusion, our multi-step strategy for identifying EV-based protein biomarkers for NAFLD provides a comprehensive approach that can also be applied to other diseases. The protein candidates identified through this approach could have significant implications for the development of non-invasive diagnostic tests for NAFLD and improve the management and treatment of this prevalent liver disorder.

## 1. Introduction

Non-alcoholic fatty liver disease (NAFLD) is the most common liver disorder globally, with a prevalence of up to 25% in the general population [1]. It is characterized by hepatic steatosis in the absence of significant alcohol intake. NAFLD ranges from non-alcoholic fatty liver (NAFL) to non-alcoholic steatohepatitis (NASH) and liver fibrosis or cirrhosis, with the latter having the highest risk of developing liver-related morbidity and mortality [1]. Therefore, due to its prevalence and potential severity, there is a need for biomarkers that can accurately diagnose, predict disease progression, and monitor response to therapy, particularly for patients with NAFL who are at risk of developing NASH. Liver biopsy has traditionally been the gold standard for the diagnosis and staging of NAFLD, but its invasiveness, cost, and potential complications have led to a search for non-invasive biomarkers [2]. Based on liver biopsy analysis, the grading and staging of NAFLD are crucial for the diagnosis, management, and prognosis of the disease. The grading system classifies the severity of the histological features of NAFLD, including steatosis, ballooning, and inflammation. The staging system assesses the extent of liver fibrosis, the most important prognostic factor in NAFLD [3,4,5,6] (Figure 1).

A variety of biomarkers have been proposed. However, none has been universally accepted as a reliable diagnostic tool for NAFLD [7]. Biomarkers for NAFLD can be broadly categorized into three types: (**I**) imaging biomarkers, (**II**) circulating biomarkers, and (**III**) genetic biomarkers. Imaging biomarkers, such as liver fat percentage by Magnetic Resonance Imaging (MRI) using proton density fat fraction (PDFF), liver stiffness measurement and magnetic resonance elastography, have shown promise in the diagnosis and staging of NAFLD [8]. However, these methods are costly and not widely available, limiting their widespread use.

Circulating biomarkers have been extensively studied in NAFLD, with many potential candidates identified. Nevertheless, these biomarkers have limitations in relation to their sensitivity/specificity, predictive value, and standardization. Although widely used and standardized worldwide, the circulating biomarkers Alanine aminotransferase (ALT), Aspartate aminotransferase (AST), and Gamma-glutamyl transferase (GGT) have limitations in their ability to specifically diagnose NAFLD [9]. These biomarkers can indicate liver dysfunction but cannot differentiate between various liver diseases or specifically identify NAFLD.

Fatty acid binding protein 4 (FABP4), cytokeratin 18 (CK-18), and fibroblast growth factor 21 (FGF21) have all emerged as potential circulating biomarkers for NAFLD. While they offer improved specificity compared with traditional liver enzymes, they have limitations that need to be addressed. These biomarkers cannot differentiate between different types of liver damage and can be elevated in other conditions that share risk factors with NAFLD [8]. Furthermore, they are still in the research stage and lack diagnostic approval. Future studies should focus on validating these biomarkers in large cohorts, establishing their diagnostic accuracy, and developing standardized protocols for their utilization in clinical practice.

Novel biomarkers that can overcome these limitations are needed to improve the accuracy and specificity of NAFLD diagnosis and management.

Predictive algorithms, such as the fibrosis-4 (FIB-4) index, fibrotic NASH index (FNI), and Forns score, have emerged as potential tools for the diagnosis of NAFLD [10,11]. These algorithms utilize readily available clinical parameters (such as ALT, Body mass index, and AST) to predict the presence or severity of NAFLD, offering advantages in terms of cost, accessibility, and standardization. However, despite their utility, there are certain limitations that need to be addressed, including issues related to specificity and the need for further validation in large cohorts.

Circulating miRNAs have emerged as promising biomarkers in NAFLD. These small non-coding RNA molecules are released into the bloodstream and reflect the molecular changes occurring in the liver. By profiling these miRNAs, researchers can potentially diagnose NAFLD, assess disease progression, and monitor treatment responses non-invasively. The unique miRNA expression patterns associated with different stages of NAFLD offer valuable insights into underlying pathogenic mechanisms [12,13]. Genetic biomarkers have also been investigated in NAFLD, with genome-wide association studies (GWAS) identifying several loci associated with NAFLD susceptibility and progression [7]. The scope of these approaches is not related to the purpose of this review.

Extracellular vesicles (EVs) are a heterogeneous population of vesicles that differ in size, content, and function [14]. They are classified based on their biogenesis and size into three main categories: exosomes (30–150 nm), microvesicles (150–1000 nm), and apoptotic bodies (>1000 nm) (Figure 2). EVs are secreted by various cell types, including liver cells, and can be found in various body fluids, such as blood, urine, and bile. EVs are now recognized as critical players in cell-to-cell communication, as they carry various biomolecules that can be delivered to target cells [14].

EVs’ role in NAFLD has been widely investigated [15], and several studies have suggested that EVs can be used as biomarkers for the diagnosis and monitoring of NAFLD [16]. The regulation of EV synthesis and secretion in various stages of NAFLD could be linked to each stage’s cellular stress, inflammation, and fibrotic responses. Thus, the cargo of EVs secreted by liver cells, including hepatocytes, cholangiocytes, and Kupffer cells, can reflect the liver’s pathophysiological state in NAFLD. Circulating EVs in NAFLD patients’ blood have been shown to carry specific proteins related to lipid metabolism, oxidative stress, inflammation, and fibrosis, which are hallmarks of NAFLD progression [17] (Figure 3).

However, most of the studies dedicated to finding biomolecules with diagnostic value in EVs derived from patients with NAFLD are based on classical technology approaches of EVs isolation and purification by ultracentrifugation followed by omic analysis of biomolecules (e.g., proteomics, RNAseq, etc.). These approximations are necessary to discover new possible biomolecule candidates, but they are too complex to be taken to the clinic.

New technologies, such as EV-Array [18] or Exoview, allow a quick and easy way to study the surface proteins of the EVs [19]. EV-Array is based on the technology of protein microarray and provides the opportunity to detect and phenotype small EVs from unpurified starting material in a high-throughput manner. Exoview is an innovative approach for the characterization of EVs surface proteins. Exoview uses nanoparticle tracking analysis (NTA) to measure the size and concentration of EVs and combines this with a microarray-based assay to analyse EVs surface proteins. The microarray-based assay uses antibodies immobilized on a glass slide to capture specific proteins from the EVs. The captured proteins are then detected using fluorescently labelled secondary antibodies. The resulting protein profile of the EVs can provide valuable information about the underlying biology [19]. Recent advances in flow cytometry have led to the development of high-sensitive flow cytometry (HSFC), which allows for the detection of EVs with higher sensitivity and specificity than traditional flow cytometry methods [20]. HSFC uses smaller sample volumes and specialized instrumentation to minimize background noise and increase signal detection. Another advantage of HSFC is its ability to provide information on multiple surface proteins simultaneously [21]. HSFC can detect and quantify the expression of multiple proteins on individual EVs directly in plasma without preceding purification steps, providing a more comprehensive analysis of EV populations.

In this review, our goal was to develop a list of potential EVs surface protein biomarkers that could aid in the diagnosis and monitoring of NAFLD. We summarize the current knowledge regarding EV surface proteins as potential biomarkers for NAFLD. Thus, we conducted a comprehensive review of the literature. (I) First, we reviewed a scientific bibliography that studied EV proteins as biomarkers in NAFLD, papers that have studied circulating proteins as biomarkers in NAFLD, and papers/reviews summarizing the main proteins involved in NAFLD biology. (II) Next, to identify potential candidates, we utilized the EV database Vesiclepedia.org to qualify each protein and select only those previously found in EVs (from any cell/tissue source). (III) Finally, we consulted the Human Protein Atlas (https://www.proteinatlas.org/), accessed on 1 April 2023, to search for the localization of each protein. We focused on membrane proteins. Finally, we categorized the list according to the biological function and/or technical applications in which these EVs proteins were involved. We have generated Table 1 with 69 protein candidates, which could be used as EVs surface biomarkers in NAFLD. Thus, Table 1 shows both surface proteins found by direct corroboration in EVs derived from key tissues in NAFLD and surface proteins that could potentially be found in EVs derived from NAFLD key tissues (Table 1, column 4). 

## 2. Novel Proposed EV Protein Biomarkers

In this category, we included EV surface proteins already proposed as NAFLD biomarkers by different studies.

**Vanin-1** is an amidohydrolase that hydrolyzes specifically one of the carboamide linkages in D-pantetheine, thus recycling pantothenic acid (vitamin B5) and releasing cysteamine. In a mice study, Povero et al. [23] showed that Lipid-Induced toxicity stimulates hepatocytes to release angiogenic EVs that require Vanin-1 on the surface for uptake by endothelial cells. Their data identify hepatocyte-derived EVs as critical signals contributing to angiogenesis and liver damage in steatohepatitis. Motomura et al. [22], using mice and human in vitro approaches, showed vanin-1 upregulation and lipid accumulation in hepatocytes in response to a high-fat diet and free fatty acids (FFA).

**TREM2.** Triggering receptor expressed on myeloid cells 2 (TREM2) is a membrane protein that forms a receptor signaling complex with the TYRO protein tyrosine kinase binding protein. Soluble TREM2 levels in blood could be a circulating marker of NAFLD [24]. TREM2 has been found in EVs derived from human neurons [26]. Additionally, TREM2 regulates EVs secretion in liver macrophages during NAFLD progression [25].

**ADAMTSL2**. ADAMTS like 2 (ADAMTSL2) is “a disintegrin and metalloproteinase with thrombospondin motifs”. It has been described as a soluble biomarker to indicate NAFLD to NASH progression in patients [27].

**IL13RA1**. Interleukin 13 receptor subunit alpha 1 (IL13RA1) is a subunit of the interleukin 13 receptor. Povero et al. [28] showed upregulated levels of IL13RA1 in cirrhotic NASH circulating EVs versus healthy controls. In the same study [28], authors found increased EV levels of **IL27RA**, **ICAM2** and **STK16** in cirrotic NASH samples.

## 3. Metabolism Related Proteins

Metabolism-related proteins are critical for the regulation of lipid metabolism, glucose homeostasis, insulin signaling, and inflammatory responses, all of which are disrupted in NAFLD.

**CD36** is a multifunctional glycoprotein that acts as a receptor/transporter for a broad range of ligands. CD36 increases FFA uptake, and in the liver, it drives hepatosteatosis onset. Clinical studies have reinforced the significance of CD36 by showing increased content in the liver of NAFLD patients [30]. Interestingly, circulating levels of a soluble CD36 (sCD36) are abnormally elevated in NAFLD patients [30]. Moreover, we showed that CD36 is expressed on circulating EVs surface and is related to the delivery of FFA from blood flow to the heart [29]. Bariatric surgery resulted in significantly altered levels of CD36 in circulating EVs of monocyte and endothelial origin [111].

**TM4SF5.** Transmembrane 4 L six family member 5 (TM4SF5) is a tetraspanin involved in nonalcoholic steatosis and further aggravation of liver disease [33]. TM4SF5 is present on the membranes of different organelles or EVs [31] and cooperates with transporters for fatty acids, amino acids, and monocarbohydrates, thus regulating nutrient uptake into hepatocytes. TM4SF5 can remodel the immune environment by interacting with immune cells during TM4SF5-mediated chronic liver diseases [32]. Interestingly, liver-derived EVs with TM4SF5 on their surface target brown adipose tissue for homeostatic glucose clearance [31].

**TM6SF2**. Transmembrane 6 superfamily member 2 (TM6SF2) encodes for a protein of undetermined function. Genetic studies have reported the association between TM6SF2 variants with hepatic triglyceride content and its impact on NAFLD [34,35].

**SLC27A5**. Bile acyl–coenzyme A synthetase (Solute carrier family 27 members 5, also known as FATP5) is an isozyme of very long-chain acyl-CoA synthetase. It is considered a hepatocyte-specific marker in circulating EVs [28]. It has been described as a functional association between SLC27A5 and TM4SF4 in fatty acids overload hepatocytes [33]. Expression of SLC27A5 is up-regulated in fat-laden hepatocytes and down-regulated during the progression from NASH to cirrhosis, likely due to fat loss occurring during the late stage of the disease. In patients, median levels of SLC27A5-positive circulating EVs were 3-4-fold greater in subjects with NASH compared with healthy controls [28].

**SGMS1.** Sphingomyelin synthase 1 (SGMS1) catalyzes the reversible transfer of phosphocholine moiety in sphingomyelin biosynthesis. Glucosylceramide (GluCer) accelerates liver steatosis, steatohepatitis, and tumorigenesis [36]. Glucosylceramide stimulates transforming growth factor beta 1 (TGFβ1) activation, which mediates liver fibrosis. Human NASH patients were shown to have higher liver GluCer synthase and higher plasma GluCer levels [36].

**GLUT1**. Solute carrier family 2 member 1 (SLC2A1) is a facilitative glucose transporter responsible for constitutive or basal glucose uptake. In NASH patients, increased liver GLUT1 levels correlate with a higher degree of steatosis [37]. Lu Wan et al. showed that EVs derived from hepatic stellate cells contain GLUT1, which is related to the metabolic switch of liver nonparenchymal cells [38].

**GLUT2**. Solute carrier family 2 member 2 (SLC2A2) facilitative hexose transporter that mediates the transport of glucose, fructose, and galactose. In NAFLD, impaired glucose uptake by the liver is mediated by GLUT2 interference [39].

**GLUT5**. Solute carrier family 2 member 5 (SLC2A5) functions as a fructose transporter that has only low activity with other monosaccharides. Increased liver levels of GLUT5 were observed in NAFLD induced by high fructose intake in rats [40].

**GLUT4**. Solute carrier family 2 member 4 (SLC2A4) is an insulin-regulated facilitative glucose transporter, which plays a key role in removing glucose from circulation. Altered liver levels of GLUT4 have been found in patients with chronic liver disease [42]. We previously demonstrated that EVs derived from cardiomyocytes have GLUT4 on their surface with functional consequences for the cardio-endothelial communication axis [41].

**AGTR1**. Angiotensin II receptor type 1 (AGTR1) is the receptor for angiotensin II, a vasoconstricting peptide. Hepatic mRNA expression of Agtr1a was downregulated in NAFLD-induced rats [45]. Moreover, gene variants of AGTR1 have been related to a predisposition to develop NAFLD in patients [43,44]. Bansal et al. showed AGTR1 presence in circulating EVs from patients with SARS-CoV-2 infection [112].

**CAV1**. Caveolin 1 (CAV1) is the main component of the caveolae plasma membranes found in most cell types. CAV1 is a crucial regulator of lipid accumulation and metabolism. It is known that hepatocytes CAV1 modulate metabolic gene profiles and function in NAFLD [46]. Increased levels of circulating EVs expressing CAV1 have been reported in melanoma patients [47].

## 4. Inflammation and Fibrosis

Inflammation and fibrosis are integral processes in the development and progression of NAFLD [113]. Inflammation in the liver is often caused by hepatocyte fat accumulation, which releases pro-inflammatory cytokines. These cytokines can damage liver cells and trigger an inflammatory response, which can lead to the development of fibrosis [114]. The persistent inflammatory response contributes to hepatocyte injury, insulin resistance, and the recruitment of immune cells, amplifying liver damage. Simultaneously, fibrosis disrupts the liver architecture, impairing liver function and ultimately leading to cirrhosis and hepatocellular carcinoma (HCC) [114].

**CXCL10.** C-X-C motif chemokine ligand 10 (CXCL10) is a pro-inflammatory cytokine that is involved in a wide variety of processes such as chemotaxis, differentiation, and activation of peripheral immune cells, regulation of cell growth, apoptosis, and modulation of angiostatic effects. During hepatocyte lipotoxicity, activated mixed lineage kinase 3 (MLK3) induces the release of CXCL10-bearing vesicles from hepatocytes, which are chemotactic for macrophages [48]. In diabetes, pancreatic beta cells in a pro-inflammatory environment release EVs with CXCL10 on the surface, which induces failure of neighbouring beta cells through activation of the CXCL10/CXCR3 axis [49].

**TGFB1**. Transforming growth factor beta 1 (TGFB1) is a secreted ligand of the TGF-beta (transforming growth factor-beta) superfamily of proteins. TGFB1 promotes hepatic stellate cell (HSC) activation and extracellular matrix production (ECM), which further contributes to the progression of NAFLD [50]. Interestingly, it has been shown that TGFB1 can be delivered on the surface of mast cell-derived EVs [51].

**TGFB2, TGFB3** and their receptors (**TGFBR1, TGFBR2, TGFBR3**). Transforming growth factor beta 2 and 3 (TGFB1 and TGFB3) are secreted ligands of proteins’ TGF-beta (transforming growth factor-beta) superfamily. Different studies have investigated the implications of these proteins in NAFLD and its progression to NASH in relation to inflammation and fibrosis [53,54], which is why it is well established that regulation of the TGFB pathway is related to liver pathology. Moreover, Rodrigues-Junior et al. wrote/published a complete review of TGFB components on EVs surface [52].

**ITGB1**. Integrin subunit beta 1 (ITGB1). Beta-1 integrins recognize the sequence R-G-D in a wide array of ligands. NASH patients’ livers have increased mRNA expression levels of ITGB1 compared with NAFLD and healthy control [55]. ITGB1 is released in EVs from hepatocytes under lipotoxic stress in mice and mediates monocyte adhesion to liver sinusoidal endothelial cells to promote hepatic inflammation [56].

**TLR2.** Toll-like receptor 2 (TLR2). Toll-like receptor (TLR) family plays a fundamental role in pathogen recognition and activation of innate immunity. TLR2 and palmitic acid cooperatively activate the inflammasome in Kupffer cells/macrophages in the development of NASH [57].

**TLR4.** Toll-like receptor 4 (TLR4) expression is upregulated in NASH patients, compared with those with NAFLD [58]. Moreover, activation of the TLR4 inflammatory pathway contributes to NAFLD severity and NASH/hepatic fibrosis [59]. Zhang et al. showed evidence indicating TLR4 transference by EVs between dendritic cells [60].

**P2RX7**. Purinergic receptor P2RX7 (P2RX7) is a receptor for ATP that acts as a ligand-gated ion channel that is responsible for ATP-dependent lysis of macrophages through the formation of membrane pores permeable to large molecules. P2RX7 regulation is related to the pathophysiological events leading to NAFLD and its inflammatory and fibrotic evolution [61].

**P2RY14**. Purinergic receptor P2Y14 (P2RY14) is a receptor for UDP-glucose and other UDP-sugar coupled to G-proteins. P2RY14 links hepatocyte death to hepatic stellate cell activation and fibrogenesis in the liver [62].

**CXCR4**. C-X-C motif chemokine receptor 4 (CXCR4) is a G-protein coupled receptor that transduces a signal by increasing intracellular calcium ion levels and enhancing MAPK1/MAPK3 activation. CXCR4 and its ligand are functionally and mechanistically involved in the progression of liver fibrosis [63,65]. Horizontal transference of CXCR4 by EVs promotes hepatocarcinoma cell migration, invasion, and lymphangiogenesis [64].

**PDGFRB and PDGFRA**. Platelet-derived growth factor receptors beta and alpha (PDGFRB and PDGFRA) are tyrosine-protein kinases that act as cell-surface receptors. In NAFLD patients, circulating levels of PDGFRB are progressively enhanced with increasing fibrosis stage. The largest difference was observed in patients with significant fibrosis, compared with no or mild fibrosis [66]. Moreover, increased expression of liver levels of PDGFRA has been associated with NAFLD to NASH progression in patients [36].

**SPHK1**. Sphingosine kinase 1 (SPHK1) catalyzes the phosphorylation of sphingosine to form sphingosine 1-phosphate (SPP), a lipid mediator with both intra- and extracellular functions. SPHK1 mediates hepatic inflammation in HFD mice and initiates proinflammatory signalling in hepatocytes [67]. SPHK1 has been found in EVs derived from the hepatocellular carcinoma cell line SK-Hep1 [68].

**ITGA1**. Integrin subunit alpha 1 (ITGA1) is a receptor for laminin and collagen. Hepatic insulin resistance associated with increased levels of liver collagen and elevated expression of ITGA1 in hepatocytes isolated from high fat (HF)-fed mice has been shown [69]. Moreover, ITGA1 facilitates hepatic insulin action while promoting lipid accumulation in mice under an HF diet.

**ACKR1**. Atypical chemokine receptor 1 (ACKR1) controls chemokine levels and localization via high-affinity chemokine binding uncoupled from classic ligand-driven signal transduction cascades. Transcriptome analysis of more than 100,000 single human cells revealed distinct endothelial subpopulations that inhabit the liver fibrotic niche. These endothelial cells express ACKR1+, restricted to cirrhotic liver tissue and induce the transmigration of leucocytes [72]. ACKR1 knockdown attenuated leucocyte recruitment by cirrhotic endothelial cells [71]. Importantly, ACKR1 has been found in EVs derived from endothelial cells [70].

**GPNMB**. Glycoprotein nmb (GPNMB) is a type I transmembrane glycoprotein. GPNMB overexpression ameliorated liver fat accumulation and fibrosis in diet-induced obesity in mice [73]. In patients with non-alcoholic steatohepatitis, serum soluble GPNMB concentrations were higher than those with simple steatosis [73].

**SCARB1**. Scavenger receptor class B member 1 (SCARB1) is a receptor for ligands such as phospholipids, cholesterol ester, lipoproteins, phosphatidylserine, and apoptotic cells. In diet-induced obesity in mice, SCARB1 deficiency increased inflammatory dyslipidaemia and adipocytes hypertrophy and attenuated hepatic steatosis [74]. Angeloni et al. demonstrate evidence of SCARB1 in EVs derived from prostate cancer cells [74].

**FGFR1**. Fibroblast growth factor receptor 1 (FGFR1) is a tyrosine-protein kinase that acts as a cell-surface receptor for fibroblast growth factors. FGFR1 is a central player in the response to liver injury and fibrosis [75].

**TNFSF10**. TNF superfamily member 10 (TNFSF10) is a cytokine that belongs to the tumor necrosis factor (TNF) ligand family. Liver expression of TNFSF10 is increased in both human and experimental NASH, and fatty murine livers are sensitized to TNFS10F-mediated hepatocyte apoptosis [76].

**CD68** is a well-established marker for identifying macrophages. Increased liver infiltration with CD68+ macrophages is related to liver fibrosis [77]. In addition, CD68 expression has been found on circulating EVs in hypertensive rats [78].

## 5. NOTCH Pathway

The Notch pathway is a conserved ligand-receptor signaling mechanism involved in the regulation of tissue homeostasis and the maintenance of stem cells in adults and normal vasculature development and angiogenesis [81]. Notch signaling is activated when specific transmembrane Notch ligands of the Jagged or Delta-like type, found on neighboring cells, engage with the extracellular domain of the receptor by proximity. Four Notch receptors have been identified in mammalian cells, **Notch1-4**, activated by five canonical ligands, Delta-like 1, 3, and 4 (**Dll1**, **3**, and **4**) and **Jagged1** and **Jagged2** [82,83]. Several studies have shown Notch signaling pathway plays a pivotal role in the regulation of NAFLD progression, from lipid accumulation to fibrosis and cancer [84]. It is well established that all NOTCH components (ligands and receptors) are functionally found in EVs from different sources [79,80].

## 6. Wnt/β-Catenin Pathway

The Wnt/β-catenin pathway is vital in embryonic development, tissue homeostasis, and cell fate determination. Dysregulation of this pathway has been implicated in numerous human diseases, including cancer and metabolic disorders [115]. There exist 19 secreted Wnt ligands coupled to 15 receptors and co-receptors. The canonical Wnt/β-catenin pathway is initiated by binding Wnt ligands to the Frizzled (FZD) receptors and co-receptors, such as LRP5/6. This interaction triggers a cascade of intracellular signaling events, leading to the stabilization and accumulation of β-catenin in the cytoplasm and posterior nucleus translocation to regulate gene expression [115]. In liver physiology, the Wnt/β-catenin pathway plays a critical role to establish and maintain liver zonation [116,117].

Emerging evidence suggests that dysregulation of the Wnt/β-catenin pathway contributes to the development and progression of NAFLD [90]. In the context of NAFLD, aberrant activation of the pathway has been observed in hepatocytes, leading to the accumulation of lipids and the development of hepatic steatosis [90]. Additionally, increased β-catenin signaling has been associated with the activation of hepatic stellate cells and the subsequent progression to fibrosis in NASH. Finally, there is increased evidence showing that different cell types secrete Wnt/β-catenin components in EVs [60,93,118]. Here, we highlight the ligands, receptors, and co-receptors directly linked to NAFLD pathology.

Using a mouse model of methionine-choline-deficient diet (MCDD)-induced NASH, Zhu et al. investigated the Wnt signalling pathways in relation to hepatic glucose oxidation [85]. While liver expression of **Wnt1** was unchanged, **Wnt3a** was significantly reduced in NASH. In contrast, expression of **Wnt5a** and **Wnt11** were increased 3-fold and 20-fold, respectively. Other mice diet-induced obese models of NASH identified increased expression of **Wnt2** and **RSPO3** (a Wnt pathway co-receptor) in liver pericentral endothelial cells [86]. Xiong et al. performed single-cell RNA sequencing on non-parenchymal cells isolated from healthy and NASH mouse livers. In that study, secretome gene analysis revealed a highly connected network of intrahepatic signaling and disruption of vascular signaling in NASH. They found altered expression levels of **Wnt9b**, **Wnt2**, **Wnt4** and **Rspo3** [87].

Interestingly, Scavo et al. found that in circulating EVs, **FZD7** levels were modulated by lifestyle interventions in patients with NAFLD [88]. In another study, Saponara et al. showed that in mice, loss of hepatic Wnt/β-catenin activity by **Lgr4/5** deletion led to loss of bile acid secretion, cholestatic features, altered lipid homeostasis, and deregulation of lipoprotein pathways, promoting NAFLD [89]. Mutations in ***LRP6*** are one of the major causes of NAFLD induction. Moreover, deprived levels of **Wnt1** and higher levels of **DKK-1**, an inhibitor of the Wnt pathway in plasma, are correlated with a raised risk of hyperlipidemia by antagonizing **LRP6** [90]. Finally, different studies have shown **β-catenin** transference between cells mediated by EVs. This implies an alternative new level of regulation for the pathway since non-ligand-receptor interaction is required for the activation [91,92,93].

## 7. Plasma/Serum Secreted Proteins in NAFLD

Recent studies have employed proteomic techniques to find plasma and serum-secreted proteins differentially expressed in patients with NAFLD compared with healthy individuals to identify new circulating biomarkers. These proteins include markers of liver damage, inflammation, oxidative stress, and metabolic dysfunction.

In a study with NAFLD and NASH patients, Niu et al. identified deregulated plasma levels of **AFM** and **PIGR** [94]. **AFM**. Afamin (AFM) is a carrier for hydrophobic molecules in body fluids. It is essential for the solubility and activity of Wnt ligands. Different groups have identified circulating levels of AFM in patients as predictive biomarkers for NAFLD development [94,95]. **PIGR**. Polymeric immunoglobulin receptor (PIGR) mediates selective transcytosis of polymeric IgA and IgM across mucosal epithelial cells. A corroboration of Niu et al. results [94] was obtained by Veyel et al. using a mice NASH model where they found increased circulating levels of PIGR associated with NASH phenotype [98]. PIGR was directly detected in EVs related to cancer biology [96,97].

Using a proteomic approach in plasma and serum samples from NAFLD patients, Corey et al. identified circulating levels of formimidoyltransferase cyclodeaminase (**FTCD**) as an indicative biomarker for the NAFLD to NASH progression [27]. **FTCD** is a folate-dependent enzyme that displays both transferase and deaminase activity.

## 8. Inter-Tissue Crosstalk

Inter-tissue crosstalk communication plays a fundamental role in the pathogenesis of NAFLD [43]. The intricate interactions between the liver, adipose tissue, skeletal muscle, and the gut contribute to the development and progression of hepatic steatosis, inflammation, and fibrosis. Excessive adiposity, particularly visceral adipose tissue, is strongly associated with NAFLD development and progression. Adipose tissue secretes various adipokines, EVs, cytokines, and FFAs that directly or indirectly influence hepatic lipid metabolism, insulin sensitivity, inflammation, and fibrosis [119]. Skeletal muscle also plays a crucial role in NAFLD. Insulin resistance, a hallmark of NAFLD, is strongly linked to skeletal muscle dysfunction. Impaired glucose and lipid uptake by skeletal muscle led to increased delivery of substrates to the liver, exacerbating hepatic lipid accumulation. Furthermore, skeletal muscle-derived myokines, such as interleukin-6 (IL-6), irisin, and myostatin, have emerged as important regulators of hepatic lipid metabolism and inflammation [120]. This section lists proteins related to inter-tissue crosstalk linked to liver pathology.

**CD40** is a member of the TNF-receptor superfamily. Its suppression is related to adipose tissue inflammation, while the absence of CD40 aggravated metabolic dysfunction in mice. CD40 expressing CD11c+ dendritic cells contribute to liver inflammation during NASH but are protective against metabolic syndrome via induction of regulatory T cells [99]. Interestingly, it has been shown that CD40 function is related to adipokines secretion [100].

**LRG1**. Leucine-rich alpha-2-glycoprotein 1 (LRG1) belongs to the leucine-reach repeat proteins family. LRG1 is an adipokine that mediates obesity-induced hepatosteatosis and insulin resistance [101]. While LRG1 is not a membrane protein, it has been found on circulating EVs surface from cancer patients [102].

**FNDC5**. Fibronectin type III domain containing 5 (FNDC5), called irisin, is a secreted myokin released from muscle cells during exercise. Several studies have emphasized that obesity is closely related to a disorder of serum irisin. Clinical data have indicated that serum irisin levels are reduced in patients with obesity-related NAFLD [104]. Exercise induces increased irisin in circulating EVs in mice and humans [103].

**AKR1B7**. Aldo-keto reductase family 1 member B7 (AKR1B7) catalyzes the NADPH-dependent reduction of various carbonyl-containing compounds to their corresponding alcohols. AKR1B7 in EVs derived from metabolically stressed adipocytes induces NASH in mice [105].

**FASN**. Fatty acid synthetase (FASN) is a multifunctional enzyme that catalyzes the de novo biosynthesis of long-chain saturated fatty acids starting from acetyl-CoA and malonyl-CoA in the presence of NADPH. FASN drives de novo lipogenesis and mediates pro-inflammatory and fibrogenic signalling in NAFLD [106]. FASN was selectively enriched in EVs derived from adipocytes under the hypoxic condition, which may increase lipid accumulation in recipient adipocytes and preadipocytes [107].

**FABP4**. Fatty acid binding protein (FABP4) is an adipose tissue-secreted adipokine implicated in the regulation of energetic metabolism and inflammation [109]. High levels of circulating FABP4 have been described in people with obesity, atherogenic dyslipidemia, diabetes, metabolic syndrome, and NAFLD. Additionally, emerging data from preclinical studies propose FABP4 as a causal actor involved in the disease progression rather than a mere biomarker for the disease [109]. Preliminary evidence from flow cytometric analyses showed that circulating EVs contain the adipocyte markers FABP4 [108] in human plasma. This implies that FABP4 was detected on the circulating EVs surface. Interestingly, adiponectin has been found in circulating EVs in mice [110].

## 9. Perspective

EVs have emerged as potential biomarkers for NAFLD due to their involvement in the development and progression of these conditions. EVs have been implicated in several processes associated with NAFLD, such as lipid metabolism, inflammation, and fibrosis. The unique composition of EV surface proteins makes them attractive candidates in biomarker discovery. One of the key advantages of EVs as biomarkers is their ability to circulate in the bloodstream, providing a minimally invasive way to access disease-related information from the liver. However, the detection and analysis of EVs in blood samples pose significant challenges, as they are heterogeneous and present at low concentrations. Therefore, identifying specific surface proteins on EVs associated with NAFLD could enhance their diagnostic potential. In this regard, circulating EVs have a cell-specific origin, which can be reflected in their expression of EV surface proteins. This feature implies the possibility to study specific circulating EVs subpopulations to bring more specific results. For example, we demonstrated that bariatric surgery resulted in significantly altered levels of CD36 in circulating EVs derived from monocyte and endothelial cells [111].

Traditional methods for EV protein analysis, such as Western Blot, mass spectrometry, SomaScan platform, or ELISA, have limited throughput and specificity. However, emerging technologies like EV-Array, Exoview or high-sensitive flow cytometry enable high-throughput analysis and multiplex detection of EVs, allowing simultaneous evaluation of multiple surface proteins. These technologies could boost the translation of EV-based biomarkers into clinical practice. Moreover, the development of novel technologies, such as the use of microfluidics or nanomaterials, could further improve the sensitivity and specificity of EV detection.

The identification and analysis of surface proteins on EVs offer a promising avenue for the development of non-invasive biomarkers for NAFLD (Figure 4). These biomarkers could potentially overcome the limitations of current diagnostic approaches and provide clinicians with a rapid, cost-effective, and reliable tool to assess disease progression and response to therapy. However, further validation studies are needed to confirm EV surface proteins’ diagnostic and prognostic value in larger patient cohorts.

## Figures and Tables

**Figure 1 ijms-24-13326-f001:**
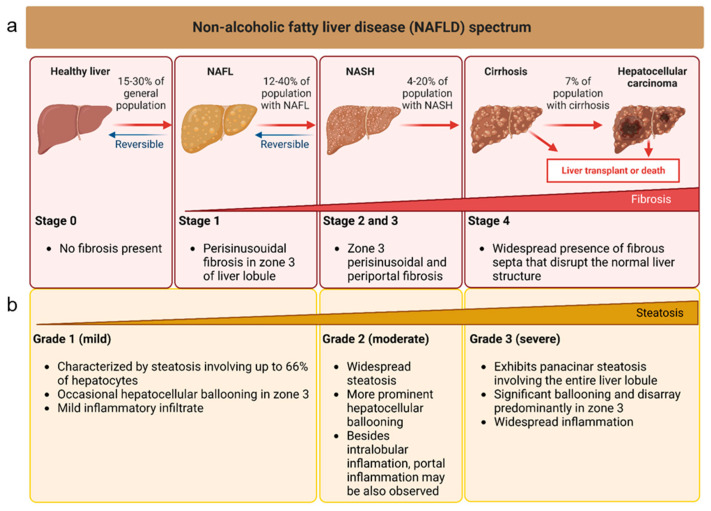
Schematic representation of the grading and staging of NAFLD based on liver biopsies analysis. (**a**) Staging is measured by the fibrosis progression. (**b**) The grade of severity is classified based on steatosis, ballooning, and inflammation.

**Figure 2 ijms-24-13326-f002:**
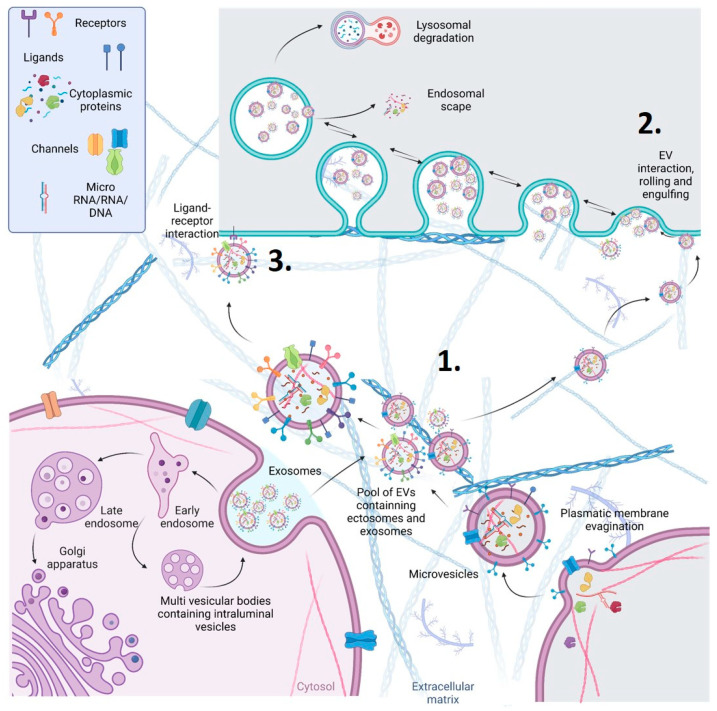
Schematic representation of the different origins and action mechanisms for EVs. **1**. EVs can be produced from the early endosomal pathway (EVs called exosomes around 30–130 nm of diameter) or by evaginations of the plasmatic membrane (EVs called microvesicles around 130–1000 nm of diameter). **2**. EVs can be engulfed by other cells through pynocitosis or phagocytosis. After being engulfed, EVs, now inside an endosome, can fuse with the membrane of the endosome and release their content (endosomal scape) degraded by fusion with lysosomes or placed again in the extracellular space (recycling). **3**. EVs can also interact directly with receptors to trigger signalling cascades in the receiving cells. Alternatively, they can directly fuse with the plasmatic membrane of the receiving cells, releasing their cytosolic cargo.

**Figure 3 ijms-24-13326-f003:**
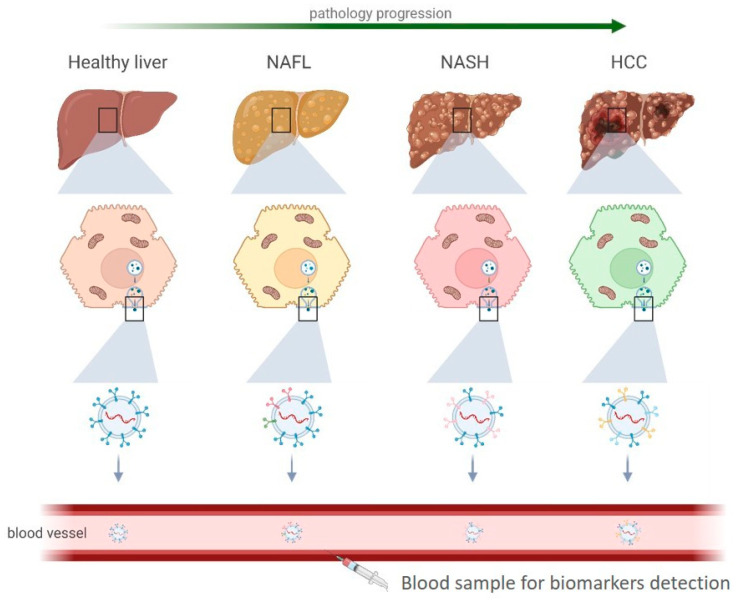
Liver cells releasing EVs to the circulation. Across different stages of NAFLD pathology, the released EVs have different patterns of surface proteins in the membrane. This differential expression of EV surface proteins could be used as NAFLD biomarkers by measuring patients’ blood samples. NAFL: Non-alcoholic fatty liver. NASH: Non-alcoholic steatohepatitis. HCC: hepatocellular carcinoma.

**Figure 4 ijms-24-13326-f004:**
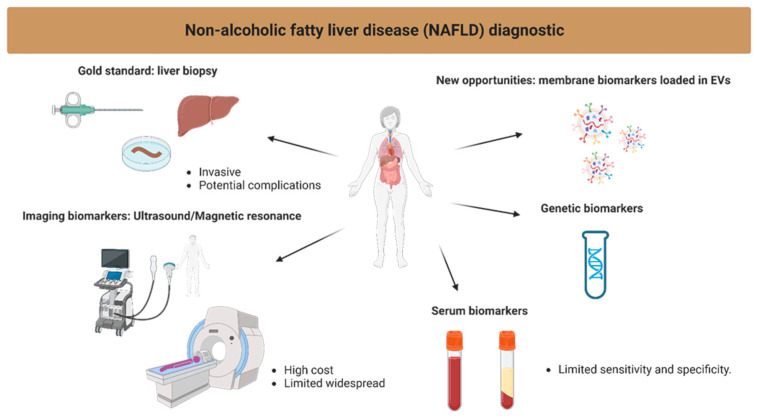
Visualization of the current methods in the diagnosis of NAFLD.

**Table 1 ijms-24-13326-t001:** EVs Protein candidates related to NAFLD.

Mechanism	Name	Description	Direct Corroboration	References
Novel proposed biomarkers	VANIN-1	Released on EVs surface from lipotoxic hepatocytes	yes	[22,23]
	TREM2	Soluble TREM2 levels correlate with NAFLD to NASH progression	no	[24,25,26]
	ADAMTS2	Soluble ADAMTS2 levels correlate with NAFLD to NASH progression	no	[27]
	IL13RA1	upregulated levels in circulating EVs in NASH	yes	[28]
	IL27RA	upregulated levels in circulating EVs in NASH	yes	[28]
	ICAM2	upregulated levels in circulating EVs in NASH	yes	[28]
	STK16	upregulated levels in circulating EVs in NASH	yes	[28]
Metabolism related proteins	CD36	circulating levels of a soluble form of CD36 are abnormally elevated in NAFLD patients	yes	[29,30]
	TM4SF5	liver-derived EVs with TM4SF5 target brown adipose tissue for glucose clearance	yes	[31,32,33]
	TM6SF2	TM6SF2 variants were related to hepatic triglyceride in NAFLD and NASH	no	[34,35]
	SLC27A5	upregulated levels in circulating EVs in NASH	yes	[28,33]
	SGMS1	NASH patients had higher liver GluCer synthase and higher plasma GluCer levels	no	[36]
	GLUT1	increased liver GLUT1 levels correlate with a higher degree of steatosis in NASH	yes	[37,38]
	GLUT2	Decreased liver levels in NAFLD	no	[39]
	GLUT5	Increased liver levels in NAFLD induced by high fructose intake in rats	no	[40,41,42]
	GLUT4	Altered liver levels in patients with chronic liver disease	no	[41,42]
	AGTR1	gene variants of AGTR1 have been related to a predisposition to develop NAFLD	no	[43,44,45]
	CAV1	hepatocytes CAV1 modulates metabolic gene profiles and function in NAFLD	no	[46,47]
Inflamation/fibrosis	CXCL10	hepatocyte lipotoxicity induces the release of CXCL10-bearing vesicles	no	[48,49]
	TGFB1	promotes HSC activation and extracellular matrix production in NAFLD	no	[50,51]
	TGFB2	drives multiple types of fibrosis during NAFLD to NASH progression	no	[52,53,54]
	TGFBR2	drives multiple types of fibrosis during NAFLD to NASH progression	no	[52,53,54]
	TGFBR3	drives multiple types of fibrosis during NAFLD to NASH progression	no	[52,53,54]
	TGFBR1	drives multiple types of fibrosis during NAFLD to NASH progression	no	[52,53,54]
	ITGB1	ITGB1 is released in EVs from hepatocytes under lipotoxic stress	yes	[55,56]
	TLR2	activate the inflammasome in Kupffer cells/macrophages in NASH development	no	[57]
	TLR4	contributes to NAFLD severity and NASH/hepatic fibrosis	no	[58,59,60]
	P2X7R	related to NAFLD and its inflammatory and fibrotic evolution	no	[61]
	P2Y14R	links hepatocyte death to hepatic stellate cell activation and fibrogenesis	no	[62]
	CXCR4	functionally and mechanistically involved in the progression of liver fibrosis	no	[63,64,65]
	PDGFRA	increased liver expression levels in NASH patients	no	[36,66]
	PDGFRB	circulating levels of PDGFRB are progressively increased with increasing fibrosis stage	no	[36,66]
	SPHK1	mediates hepatic inflammation in mice	yes	[67,68]
	ITGA1	facilitates hepatic insulin action while promoting lipid accumulation in mice	no	[69]
	ACKR1	related to leucocyte recruitment by cirrhotic endothelial cells	no	[70,71,72]
	GPNMB	increased serum levels in NASH	no	[73]
	SCARB1	SCARB1 deficiency increased inflammatory dyslipidaemia and adipocytes hypertrophy	no	[74]
	FGFR1	central player in the response to liver injury and fibrosis	no	[75]
	TNFSF10	increased liver expression levels in NASH	no	[76]
	CD68	Increased liver infiltration with CD68+ macrophages is related to liver fibrosis	yes	[77,78]
NOTCH	NOTCH1	directly involved in NAFLD development	no	[79,80,81,82,83,84]
	NOTCH2	directly involved in NAFLD development	no	[79,80,81,82,83,84]
	DLL1	directly involved in NAFLD development	no	[79,80,81,82,83,84]
	DLL3	directly involved in NAFLD development	no	[79,80,81,82,83,84]
	DLL4	directly involved in NAFLD development	no	[79,80,81,82,83,84]
	JAG1	directly involved in NAFLD development	no	[79,80,81,82,83,84]
	JAG2	directly involved in NAFLD development	no	[79,80,81,82,83,84]
WNT/β-catenin	WNT1	related to hepatic glucose oxidation in NASH	no	[85]
	WNT3a	related to hepatic glucose oxidation in NASH	no	[85]
	WNT5a	related to hepatic glucose oxidation in NASH	no	[85]
	WNT11	related to hepatic glucose oxidation in NASH	no	[85]
	WNT2	increased expression levels in liver pericentral endothelial cells in NASH	no	[86]
	RSPO3	increased expression levels in liver pericentral endothelial cells in NASH	no	[86]
	WNT9b	altered liver expression levels in NASH	no	[87]
	WNT4	altered liver expression levels in NASH	no	[87]
	FZD7	modulated levels by lifestyle intervention in NAFLD patients	yes	[88]
	LGR4/5	its activity promotes NAFLD	no	[89]
	LRP6	Mutations in LRP6 are one of the major causes of NAFLD induction	no	[90]
	DKK1	related to hyperlipidaemia in NAFLD	no	[90]
	β-catenin	efector of the pathway. It has been found in EVs	no	[91,92,93]
Plasma/serum secreted proteins in NAFLD	AFM	desregulated plasma levels in NAFLD and NASH patientis	no	[94,95]
	PIGR	desregulated plasma levels in NAFLD and NASH patientis	no	[94,96,97,98]
	FTCD	proposed indicative biomarker for NAFLD to NASH progression	no	[27]
Inter-tissue crosstalk	CD40	CD40 expressing CD11c+ dendritic cells contribute to liver inflammation in NASH	no	[99,100]
	LRG1	adipokine that mediates obesity-induced hepatosteatosis and insulin resistance	no	[101,102]
	FNDC5	serum irisin levels are reduced in patients with obesity-related NAFLD	yes	[103,104]
	AKR1B7	AKR1B7 in EVs derived from metabolic stressed adipocytes induce NASH in mice	yes	[105]
	FASN	drives de novo lipogenesis, inflammation and fibrogenic signalling in NAFLD	yes	[106,107]
	FABP4	High levels of circulating FABP4 have been described in NAFLD patients	yes	[108,109,110]

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
