# Peer review of "Comprehensive Strategy for Identifying Extracellular Vesicle Surface Proteins as Biomarkers for Non-Alcoholic Fatty Liver Disease"

_ijms, 2023, doi:10.3390/ijms241713326_

Round 1

Reviewer 1 Report

The manuscript by Garcia et al identifies and discusses potential biomarkers aimed at the diagnosis of NAFLD using extracellular vesicles. The manuscript well-written and the figures and tables contribute to the understanding of the information. The following aspects need to be addressed before publication:  

Major:

1. In general, several entries of the table (e.g., AGTR1, P2X7R, P2Y14R, ITGA1, FGFR1, TNFSF10, NOTCH, DLL, JAG, WNT, RSPO3, LGR4/5, LRP1, DKK, GLUT2, 4 and 5) are not supported by appropriate references demonstrating the presence of these markers in either liver-derived EVs or plasma EVs. In some cases, evidence is based on rather unrelated models (e.g., cardiomyocyte EVs, breast cancer EVs, prostate cancer EVs, etc.). The authors should clearly distinguish those potential markers for which the relevance in NAFLD and the presence in liver-derived or plasma-EVs has been clearly demonstrated and those candidates, which might be relevant but still have not been demonstrated in the right population of EVs. 

2. Similarly, in the discussion in sections 2 to 8, the authors should improve the distinction between proteins found in the EVs with a better biomarker potential than others which have only been described in unrelated populations of EVs or unrelated cell culture models. 

3. Lines 45-50 and Figure 1. While the classification described by the authors in the figure 1 is based on liver biopsy analysis, the reference provided (Ref. Nr. 3) describes rather non-invasive methods to estimate the disease burden. Additional references should be provided to cover all aspects presented in the figure. 

4. Figure 2. The distinction between steps 3 and 4 is not fully clear. In fact, only the interaction EV-target cell through membrane receptors is shown. 

5. Table 1. Reference 105 shows the presence of adiponectin in EVs, not FABP4 as stated in the table. Please, clarify. 

6. The authors highlight the advantages of using novel methods for EV analysis such as the ExoView or the HSFC. Was any of the proteins proposed as biomarkers analyzed or detected by any of these methods?

Minor:

7. Figure 3. The sentence “Blood sample from biomarkers detection” should be reformulated to “Blood sample for biomarker detection”. In addition, the abbreviations should be defined in the figure legend. 

8. Line 159. Please, provide a reference for the Human Protein Atlas.

9. Table 1. Second row. Please, replace “TREM2 levels correlates” by “TREM2 levels correlate”.

Author Response

Dear Reviewer 1,

thanks a lot for your comments and suggestion. We tried to address all your points and we are sure that our manuscript has improved after your revision. We hope you like this new improved version. Please find below the answers to your comments. We had attached the new manuscript version with the tracking changes "ON" so you can easily take a look at our corrections.

Best wishes

Major:

  1. In general, several entries of the table (e.g., AGTR1, P2X7R, P2Y14R, ITGA1, FGFR1, TNFSF10, NOTCH, DLL, JAG, WNT, RSPO3, LGR4/5, LRP1, DKK, GLUT2, 4 and 5) are not supported by appropriate references demonstrating the presence of these markers in either liver-derived EVs or plasma EVs. In some cases, evidence is based on rather unrelated models (e.g., cardiomyocyte EVs, breast cancer EVs, prostate cancer EVs, etc.). The authors should clearly distinguish those potential markers for which the relevance in NAFLD and the presence in liver-derived or plasma-EVs has been clearly demonstrated and those candidates, which might be relevant but still have not been demonstrated in the right population of EVs. 

Thanks for the comment, it is a great point and we totally agree. It is difficult to distinguish between surface proteins already found in EVs from key tissues and others that could POTENTIALLY be found in EVs derived from key tissues. To address this major issue, we included a new column in Table 1 and an explanation in the introduction (page 5) to clarify it.

  1. Similarly, in the discussion in sections 2 to 8, the authors should improve the distinction between proteins found in the EVs with a better biomarker potential than others which have only been described in unrelated populations of EVs or unrelated cell culture models. 

Thanks, we solve this point with the above approach (point 1).

  1. Lines 45-50 and Figure 1. While the classification described by the authors in the figure 1 is based on liver biopsy analysis, the reference provided (Ref. Nr. 3) describes rather non-invasive methods to estimate the disease burden. Additional references should be provided to cover all aspects presented in the figure. 

Thanks, and sorry about the confusion. In reference 3, Kechagias et. al made a revision for non-invasive methods, but in the introduction, they recapitulate the traditional methods. We believe that was enough information. But it can be a misunderstanding. Then, as you suggested we added extra references to better address traditional methods for NAFLD diagnosis (references 116-118).

  1. Figure 2. The distinction between steps 3 and 4 is not fully clear. In fact, only the interaction EV-target cell through membrane receptors is shown. 

Thanks for the comment. We modified Figure 2 and the figure legend to clarify it.

  1. Table 1. Reference 105 shows the presence of adiponectin in EVs, not FABP4 as stated in the table. Please, clarify. 

Thanks and sorry about the misunderstanding. We have relocated the references to clarify it. Please check page 12.

  1. The authors highlight the advantages of using novel methods for EV analysis such as the ExoView or the HSFC. Was any of the proteins proposed as biomarkers analyzed or detected by any of these methods?

This is a really good point, but we believe it exceeds the scope of the review. The aim here is to make a list of potential protein candidates to assay (further than review those technologies). Moreover, there are extensive bibliography supporting the efficacy of those approaches, which indicates that they are strong methods to detect EVs surface proteins.  For example, we used HSFC to detect CD36 in references 106 and 16. Malene Jørgensen et al., showed in 2013 the accuracy of EV-array to detect EV surface proteins (https://onlinelibrary.wiley.com/doi/10.3402/jev.v2i0.20920).

Minor:

  1. Figure 3. The sentence “Blood sample from biomarkers detection” should be reformulated to “Blood sample for biomarker detection”. In addition, the abbreviations should be defined in the figure legend. 

DONE

  1. Line 159. Please, provide a reference for the Human Protein Atlas.

DONE

  1. Table 1. Second row. Please, replace “TREM2 levels correlates” by “TREM2 levels correlate”.

DONE

Reviewer 2 Report

In this current article entitled “Comprehensive strategy for identifying extracellular vesicle surface proteins as biomarkers for Non-Alcoholic Fatty Liver Disease” by Garcia et al was devoted to understanding the use of exosomes as potential biomarkers in the diagnosis and prognosis of NAFLD. The diagnosis and treatment of NAFLD are major clinical problems, and researchers are searching desperately for novel approaches to improve the quality of life of patients and the prognosis of the disease. These tiny vesicles have only very lately been identified as a potentially fruitful route for the discovery of biomarkers in different diseases. Additionally, they highlighted the Human Protein Atlas to search for the localization of each protein. Overall, I enjoyed reading this well-written paper and have no further comments.

Author Response

Dear Reviewer 2,

thanks a lot for your words. We are glad that you enjoyed reading our review.

Best wishes

Reviewer 3 Report

Comprehensive strategy for identifying extracellular vesicle surface proteins as biomarkers for Non-Alcoholic Fatty Liver Disease

The review article aims to consolidate a comprehensive list of potential EV membrane proteins as biomarkers for NAFLD diagnosis. However, the overall design of the article needs thorough revision to make the topic more impactful.

1.       In the introduction section authors have discussed about circulating biomarkers employed for NAFLD prognosis. However, there was little information about miRNAs. As miRNAs play a key role as circulating biomarkers, authors need to include a short description of the role of miRNA in NAFLD prognosis in the introduction.

2.       Add a section about how EV synthesis is regulated in various stages of NAFLD?

3.       Authors have tried to consolidate the role of various proteins that are having a role in NAFLD. But the topic of the review is the role of extracellular vesicle surface proteins as biomarkers for NAFLD. So authors should add more information on the surface expression of these proteins in EVs during various stages of NAFLD. Most of the proteins mentioned in the review are discussed mainly based only on their role in NAFLD, not on their differential expression on the EV surface during various stages of NAFLD. So additional information from animal / human studies focusing on EV proteins in NAFLD should be added to make the topic more impactful.

4.       EVs are the key players in inter-organ cross-talk. So more information should be added on how various EV surface proteins mediate various organ cross-talk in NAFLD progression. Also, discuss about tissue-specific targets of these proteins that mediate this organ cross-talk.

5.       Add a section about the recent findings from human clinical studies exploiting EV surface markers as prognostic markers of various stages of NAFLD to emphasize the translational applicability of the topic.

English language is fine. Only minor editing of the English language is required. 

Author Response

Dear Reviewer 3,

thanks a lot for your comments and suggestion. We tried to address all your points and we are sure that our manuscript has improved after your revision. We hope you like this new improved version. Please find below the answers to your comments. We had attached the new manuscript version with the tracking changes "ON" so you can easily take a look at our corrections.

Best wishes

  1. In the introduction section authors have discussed about circulating biomarkers employed for NAFLD prognosis. However, there was little information about miRNAs. As miRNAs play a key role as circulating biomarkers, authors need to include a short description of the role of miRNA in NAFLD prognosis in the introduction.

Thanks for the comment. We agree with it. Please check page 3 to review our short description of the role of miRNAs as biomarkers in NAFLD.

  1. Add a section about how EV synthesis is regulated in various stages of NAFLD?

Great point, it is really a super interesting aspect about Evs' role in NAFLD, especially their role as biomarkers, thanks. To the best of our knowledge, there is no solid understanding about it. There are some works pointing in the direction where in the early stages of NAFLD, such as simple steatosis, an increase in lipid accumulation within hepatocytes triggers cellular stress responses which turns in an increased release of EVs, maybe contributing to local inflammation. In advanced NAFLD stages, such as fibrosis and cirrhosis, ongoing liver damage triggers the activation of stellate cells and fibrogenesis. This can result in increased secretion of EVs containing profibrotic factors, contributing to extracellular matrix remodeling and fibrosis propagation. Additionally, in the context of cirrhosis, impaired liver function may affect the synthesis and cargo composition of EVs, leading to altered intercellular communication and potentially affecting systemic processes. Overall, the regulation of EV synthesis in various stages of NAFLD could be linked to the cellular stress, inflammation, and fibrotic responses characteristic of each stage. Nevertheless, we believe that kind of information is out of the scope of the review. We want to keep the reader focused on the main idea, we preferred to encircle all that concept under the phrase (page 4) “The cargo of EVs secreted by liver cells, including hepatocytes, cholangiocytes, and Kupffer cells, can reflect the liver's pathophysiological state in NAFLD. Circulating EVs in NAFLD patients' blood have been shown to carry specific proteins that are related to lipid metabolism, oxidative stress, inflammation, and fibrosis, which are hallmarks of NAFLD progression [12] (Figure 3)”. We have added a sentence to introduce the concept of EVs synthesis and secretion across different NAFLD stages. Please check page 4 to review our correction.

  1. Authors have tried to consolidate the role of various proteins that are having a role in NAFLD. But the topic of the review is the role of extracellular vesicle surface proteins as biomarkers for NAFLD. So authors should add more information on the surface expression of these proteins in EVs during various stages of NAFLD. Most of the proteins mentioned in the review are discussed mainly based only on their role in NAFLD, not on their differential expression on the EV surface during various stages of NAFLD. So additional information from animal / human studies focusing on EV proteins in NAFLD should be added to make the topic more impactful.

Thanks for the comment and sorry about the misunderstanding. On page 6, we state “In this review, our goal was to develop a list of potential EVs surface protein biomarkers that could aid in the diagnosis and monitoring of NAFLD. We summarize the current state of knowledge regarding EV surface proteins as potential biomarkers for NAFLD”. We are not trying to consolidate the role of proteins in NAFLD. We are proposing a method to select EVs surface proteins with potential relevance in NAFLD diagnosis.

We agree with your point about adding more information on each protein would make the topic more impactful. Nevertheless, currently, there is not enough knowledge to argue about “the surface expression of these proteins in EVs during various stages of NAFLD”. Maybe, we could do it for some candidates, but not for all. Please have in mind that we are proposing our method based on the idea to apply it with high throughput technologies, as a first screening approach. 

  1. EVs are the key players in inter-organ cross-talk. So more information should be added on how various EV surface proteins mediate various organ cross-talk in NAFLD progression. Also, discuss about tissue-specific targets of these proteins that mediate this organ cross-talk.

Thanks for the comment, it is a really interesting point to argue about. Nevertheless, we believe that all the information that we provided about this topic on page 12, is enough for the purpose of this review.

  1. Add a section about the recent findings from human clinical studies exploiting EV surface markers as prognostic markers of various stages of NAFLD to emphasize the translational applicability of the topic.

Thanks for the comment, it is a key point for our review. We do not know of any clinical trial showing specific results related to our proposed approach. The only study that we could mention is “THE MULTISITE STUDY” (https://clinicaltrials.gov/study/NCT05699863?cond=NAFLD&term=Extracellular%20Vesicles&rank=1). We are the authors of this project and we are having super interesting results pointing in the direction of our review. Unfortunately, those results are not published yet. Thus, we believe it is not worth it to mention our clinical trial. Nevertheless, if you know of any clinical trial related to the aim and scope of our review please let us know and we will be happy to mention it in our manuscript.

Round 2

Reviewer 1 Report

The authors have addressed all my concerns and delivered an improved version of their manuscript. 

Reviewer 3 Report

The authors have made satisfactory revisions.